# Prognostic Values of Ferroptosis-Related Proteins ACSL4, SLC7A11, and CHAC1 in Cholangiocarcinoma

**DOI:** 10.3390/biomedicines12092091

**Published:** 2024-09-13

**Authors:** Supakan Amontailak, Attapol Titapun, Apinya Jusakul, Raynoo Thanan, Phongsaran Kimawaha, Wassana Jamnongkan, Malinee Thanee, Papitchaya Sirithawat, Anchalee Techasen

**Affiliations:** 1Medical Science Program, Faculty of Associated Medical Sciences, Khon Kaen University, Khon Kaen 40002, Thailand; thitapa_a@kkumail.com; 2Centre for Research and Development of Medical Diagnostic Laboratories (CMDL), Faculty of Associated Medical Sciences, Khon Kaen University, Khon Kaen 40002, Thailand; apinjus@kku.ac.th (A.J.); macphongsaran@gmail.com (P.K.); papitchayasiri@kkumail.com (P.S.); 3Departments of Surgery, Faculty of Medicine, Khon Kaen University, Khon Kaen 40002, Thailand; attati@kku.ac.th; 4Cholangiocarcinoma Research Institute, Khon Kaen University, Khon Kaen 40002, Thailand; wassana_jk@hotmail.co.th; 5Departments of Biochemistry, Faculty of Medicine, Khon Kaen University, Khon Kaen 40002, Thailand; raynoo@kku.ac.th; 6Departments of Pathology, Faculty of Medicine, Khon Kaen University, Khon Kaen 40002, Thailand; malitha@kku.ac.th

**Keywords:** ACSL4, SLC7A11, CHAC1, ferroptosis, cholangiocarcinoma

## Abstract

Background: The epithelial malignant tumor known as cholangiocarcinoma (CCA) is most commonly found in Southeast Asia, particularly in northeastern Thailand. Previous research has indicated that the overexpression of acyl-CoA synthetase long-chain family member 4 (ACSL4), solute carrier family 7 member 11 (SLC7A11), and ChaC glutathione-specific γ-glutamylcyclotransferase (CHAC1) as ferroptosis-related proteins is associated with poorer prognosis in several cancers. The role of these three proteins in CCA is still unclear. The present study aimed to investigate the expression levels of ACSL4, SLC7A11, and CHAC1, all potential ferroptosis biomarkers, in CCA. Methods: The ACSL4, SLC7A11, and CHAC1 protein expression levels in 137 CCA tissues were examined using immunohistochemistry, while 61 CCA serum samples were evaluated using indirect ELISA. The associations between the expression levels of ACSL4, SLC7A11, and CHAC1 and patient clinicopathological data were evaluated to determine the clinical significance of these proteins. Results: The expression levels of ACSL4, SLC7A11, and CHAC1 were assessed in CCA tissues. A significant association was observed between high ACSL4 levels and extrahepatic CCA, tumor growth type, and elevated alanine transferase (ALT). There was also a positive association between elevated SLC7A11 levels and tumor growth type. Additionally, the upregulation of CHAC1 was significantly associated with a shorter survival time in patients. High levels of ACSL4 and SLC7A11 in CCA sera were both significantly associated with advanced tumor stages and abnormal liver function test results, indicating that they could be used as a reliable prognostic biomarker panel in patients with CCA. Conclusions: The results of the present study demonstrated that the upregulation of ACSL4, SLC7A11, and CHAC1 could be used as a valuable biomarker panel for predicting prognosis parameters in CCA. Furthermore, ACSL4 and SLC7A11 could potentially serve as complementary markers for improving the accuracy of prognosis prediction when CCA sera is used. These less invasive biomarkers could facilitate effective treatment planning.

## 1. Introduction

Cholangiocarcinoma (CCA) is a type of cancerous tumor that develops from cells lining the bile ducts of the liver and biliary tract. CCA is categorized according to the location of its primary lesion into intrahepatic CCA or extrahepatic CCA (eCCA), and eCCA is further divided into perihilar or distal CCA [1]. The incidence of CCA is rare in Western countries, but it is the second most common primary hepatobiliary cancer in northeast Thailand [2]. In northeast Thailand, infections caused by the liver fluke, *Opisthorchis viverrini* (OV), have been reported as the leading cause of bile duct inflammation that leads to CCA. The early stage of CCA lacks specific clinical signs for diagnosis, whereas the late stage is more aggressive [3]. Most patients with CCA present at a hospital with late-stage disease and pass away before surgery can be performed. The survival rate of patients 2 years after diagnosis is only 8.1%, while surgical patients achieve a 5-year survival rate of 20.6% [4,5]. Therapeutic approaches for patients with CCA can vary considerably between individuals. At present, ensuring prolonged survival with comfort is quite challenging, mainly due to the generally low success rates of the currently available treatments.

To ensure an accurate prognosis and the selection of an effective treatment, it is necessary to identify a marker for predicting the prognosis of patients with CCA. A number of fields are now researching CCA biomarkers for use as diagnostic and prognostic markers or as therapeutic markers. For instance, the use of carbohydrate antigen 19-9 (CA19-9) and carcinoembryonic antigen (CEA) levels as biomarkers in the diagnosis of CCA is a specific approach that should be further studied [4]. A previous study found that CCA could be separated from non-CCA and normal cases using a Decision Tree (DT) algorithm. The DT algorithm including CA19-9 and a biomarker panel (S100 calcium-binding protein A9, mucin 5AC, transforming growth factor 1, and angiopoietin-2) showed high efficiency in the diagnosis of CCA [6]. Another study by Kimawaha et al. [7] demonstrated that alteration in glycoprotein sialylation was significantly correlated with poor prognosis in CCA. A combination of CA19-9 and glycoprotein sialylation levels and 2,6- and 2,3-sialylated glycoforms was particularly effective in distinguishing between CCA and hepatocellular carcinoma (HCC). In addition, a truncating mutation of serine/threonine kinase 11 (*STK11*) promotes tumor progression, and low expression of the STK11 protein is correlated with poor prognosis, mostly in papillary CCA. As a result, STK11 could be used as a biomarker or in targeted therapy for CCA [8]. Nevertheless, the markers identified in previous investigations showed limited specificity and sensitivity. Therefore, identifying novel molecules that can serve as biomarkers for therapy and patient assessment may enhance treatment efficacy.

Ferroptosis is a form of regulated cell death characterized by the accumulation of reactive oxygen species (ROS), which leads to the accumulation of iron and lipid peroxides. Ferroptotic cells present with a small mitochondrial size, thickening of the mitochondrial membrane, and reduced or absent mitochondrial cristae [9]. The main role of ferroptosis is to eliminate abnormal cells, such as rapidly growing cells or cancer cells. For cancer cells in particular, ferroptosis is a double-edged sword. Cancer cells require higher levels of iron than normal cells, which can cause ferroptosis. On the other hand, cancer cells need to survive and avoid the ferroptosis process. Cancer cells attempt to increase antioxidant levels to reduce ROS and lipid oxidation levels, which are associated with oxidative stress. As a result, some ferroptosis-related proteins are upregulated, and cancer cells are not eliminated. Interestingly, the upregulation of these ferroptosis-related proteins may be a weakness in cancer cells that can be targeted to provide biomarkers for therapeutic treatment [10,11]. Several studies have indicated that the imbalance of iron homeostasis and metabolism affects iron accumulation and lipid ROS accumulation in a number of cancer types [5,12]. Cancer cells require a high metabolic rate for heightened cell proliferation, which produces a high level of ROS, leading to ferroptosis. However, cancer cells do not require death; therefore, they generate antioxidant molecules against the ferroptosis process. The inhibition of these antioxidant molecules (leading to ferroptosis) may be applied as a potential therapeutic target in the context of cancer treatment [12].

Several proteins are involved in the ferroptosis mechanism, such as those associated with iron metabolism, lipid metabolism involving induction and inhibition molecules, and antioxidant mechanisms that regulate ferroptosis. These proteins include acyl-CoA synthetase long-chain family member 4 (ACSL4), solute carrier family 7 member 11 (SLC7A11), and ChaC glutathione-specific γ-glutamylcyclotransferase (CHAC1). A study that involved cell functional tests and xenografts as part of a clinical trial demonstrated that ACSL4 has an essential role in sorafenib-induced ferroptosis and may serve as a biomarker for predicting drug sensitivity in HCC [13]. SLC7A11 (also termed system Xc-) is an amino acid transporter that regulates cystine and glutamate cellular levels and is important for the antioxidant homeostasis process and ferroptosis. Another study found a correlation between SLC7A11 expression and cancer prognosis [14]. A number of compounds, such as erastin, sulfasalazine, and sorafenib, act as SLC7A11 inhibitors and have been found to induce ferroptosis, leading to cancer cell death [15]. CHAC1 degrades intracellular glutathione (GSH) and promotes the ferroptosis of tumor cells. CHAC1 is involved in the γ-glutamyl cycle and digests GSH into 5-oxoproline and Cys-Gly dipeptide, which decreases GSH levels in cells and promotes apoptosis and ferroptosis [16]. A study found that CHAC1 is a potential predictor of drug sensitivity and prognosis, as well as brain metastasis originating from primary breast cancer [17]. Additionally, the drug sensitivity and viability of prostate cancer cell lines to docetaxel were predicted by using CHAC1 as a marker [18]. Furthermore, ACSL4 and SLC7A11 were identified as potential blood biomarkers for diagnosis and for predicting prognosis, respectively [19,20].

Therefore, the present study aimed to determine the expression levels of ACSL4, SLC7A11, and CHAC1 in a CCA tissue microarray and sera. The patterns of these three biomarkers were analyzed, alongside laboratory results and clinicopathological data, to determine their potential use as predictive or prognostic biomarkers in CCA for treatment planning.

## 2. Materials and Methods

### 2.1. Patient Samples

CCA tissues (n = 137) for the microarray were obtained from patients who underwent surgery from 2005 to 2017 at Srinagarind Hospital, Khon Kaen University (Khon Kaen, Thailand), and were diagnosed with CCA by a pathologist. Sera from patients with CCA (n = 61) were also collected from individuals admitted to Srinagarind Hospital. The obtained sera were divided into two groups based on CCA staging: early CCA (stage I-II) and advanced CCA (stage III-IV) groups. The samples were stored at the Cholangiocarcinoma Research Institute (CARI) at Khon Kaen University (Khon Kaen, Thailand). The present study was reviewed by The Khon Kaen University Ethics Committee for Human Research (approval no. HE641654) and was performed in accordance with the Declaration of Helsinki and the ICH Good Clinical Practice Guidelines. Written informed consent was obtained from each patient.

### 2.2. Patient Clinicopathological and Laboratory Data

The clinicopathological characteristics and laboratory data of the patients with CCA included in the present study, including sex, age, tumor location (intrahepatic or extrahepatic), tumor growth type (intraductal, mass-forming, or mixed type), lymph node metastasis, distant metastasis, tumor staging (defined according to the American Joint Committee on Cancer Staging Manual, 8th edition) [21,22], history of OV infection, chemotherapy treatment, liver function test results, and patient survival rate, were obtained from CARI.

### 2.3. Immunohistochemistry of the Tissue Microarray

The CCA tissue microarray was constructed using a 2.0 mm diameter needle, with 70 tissue cores per block. The tissue in the paraffin blocks was sectioned using a microtome. The sections were incubated at 60 °C for 30 min, before xylene deparaffinization and subsequent rehydration using a graded ethanol series (100, 95, 80, and 70%). Antigen retrieval was performed by heating the sections in 1X sodium citrate buffer (pH 6.0) for 10 min using a microwave (for ACSL4 and SLC7A11) or for 2 min using a pressure cooker (for CHAC1), followed by cooling and washing with PBS. Then, the sections were incubated with 0.3–3% hydrogen peroxide to block endogenous peroxidase activity, followed by 5% bovine serum albumin for 30 min to block non-specific binding. Each section was then incubated with a SLC7A11 (1:1000; Abcam; cat. no. ab37185, Waltham, MA, USA), ACSL4 (1:1000; Abcam; cat. co. ab155282, Waltham, MA, USA), or CHAC1 (1:100; Proteintech Group, Inc.; cat. no. 15207-1-AP, Chicago, IL, USA) primary antibody at 4 °C overnight. Next, the sections were subjected to three washes with 1X PBS + 0.1% Tween 20 and then incubated with a horseradish peroxidase-conjugated secondary antibody (Rabbit; EnVision^TM^; Dako; Agilent Technologies, Inc. cat. no. K4003, Carpinteria, CA, USA) for 1 h at room temperature. After incubation, the sections were again washed with 1X PBS + 0.1% Tween 20. Peroxidase activity was then revealed using 3,3′-diaminobenzidine substrates (Vector Laboratories, Inc.; Maravai LifeSciences, Burlingame, CA, USA), and the sections were counterstained with Mayer’s hematoxylin (Dako; Agilent Technologies, Inc.; cat. no. S3309, Santa Clara, CA, USA). Next, the sections were dehydrated with a graded ethanol series (70, 80, 90, and 100%) and then mounted using Bio Mount HM (Bio Optica Milano SpA, Milan, Italy). Finally, the slides were examined under a light microscope (Nikon ECLIPSE Ci-L plus biological microscope, Tokyo, Japan) at magnification (×40).

The immunohistochemical results were assessed using a histoscore (H-score) [23]. The expression levels of the ACSL4, SLC7A11, and CHAC1 proteins in the cancer cells were quantified as the percentage of positively stained protein in the cytoplasm (0–100%), multiplied by the intensity of the staining. The staining was divided into four intensity types: 0 (no staining), 1 (weak), 2 (moderate), and 3 (strong). The possible scores therefore ranged from 0 to 300. The expression level was categorized as either low or high based on the median H-score for each protein as the cut-off.

### 2.4. Indirect ELISA of the CCA Sera

An indirect ELISA was performed to evaluate the diagnostic potential of ACSL4 and SLC7A11 circulating proteins as biomarkers in CCA serum. In total, 61 serum samples from patients with CCA were analyzed. The serum was diluted in coating buffer (pH of 9.6) at a dilution of 1:100; then, 100 µL of diluted serum was applied to each duplicate well of a 96-well MaxiSorp flat-bottom plate. The plate was incubated overnight and then washed with 1X PBS + 0.05% Tween 20, which was discarded. Non-specific binding was blocked with 5% skim milk in coating buffer for 1 h at 37 °C; then, the plate was again washed with 1X PBS + 0.05% Tween 20. Primary antibodies against ACSL4 (1:1000; Abcam; cat. no. ab155282) and SLC7A11 (1:1000; Abcam; cat. no. ab37185) were diluted in incubation buffer (pH 7.4); then, 100 µL was added to each well. The plate was incubated at 37 °C for 1 h, after which it was washed using 1X PBS + 0.05% Tween 20. Following washing, the plate was incubated with a horseradish peroxidase-conjugated secondary antibody (anti-rabbit IgG; Invitrogen; Thermo Fisher Scientific, Inc.; cat. no. G-21234, Waltham, MA, USA). The plate was again washed and then incubated with o-phenylenediamine dihydrochloride (OPD) substrate in the dark for 30 min. To stop the reaction, 100 µL/well 4M H_2_SO_4_ was added. The activity was then determined using an ELISA reader at an optical density (OD) of 492 nm. The protein expression level was categorized as either low or high based on the median OD value for each protein as the cut-off.

### 2.5. Statistical Analyses

The statistical analyses were performed using SPSS (version 27.0; IBM Corp., Armonk, NY, USA) and GraphPad Prism (version 8.0; Dotmatics, Boston, MA, USA) software. The continuous data are presented as the mean ± SD, and the categorical data are presented as percentages. The association between the expression of each protein and the clinicopathological and laboratory data of patients with CCA tissue and sera were evaluated using the Chi-square test and z-test. Kaplan–Meier survival analysis was used to evaluate the overall survival time (OS), and the log-rank test was used to compare survival between the low- and high-expression groups. Univariate and multivariate analyses were used to determine the variables affecting OS. For comparisons of the OD results of protein expression in sera, the Mann–Whitney U test was employed. *p* < 0.05 was considered to indicate a statistically significant difference.

## 3. Results

### 3.1. ACLS4, SLC7A11, and CHAC1 Protein Expression Levels in CCA Tissues

The complete data of 137 patients diagnosed with CCA by a pathologist at Srinagarind Hospital of Khon Kaen University were obtained. The mean age of the patients was 60.29 years (range, 39–82 years). Of the 137 patients, 86 (62.8%) were male and 52 (37.2%) were female. The patient laboratory results from Srinagarind Hospital of Khon Kaen University were obtained for analysis. Immunohistochemical analysis of the tissue microarray constructed from 137 CCA tissue samples was performed to evaluate ACLS4, SLC7A11, and CHAC1 protein expression. The immunoreactivity of ACSL4, SLC7A11, and CHAC1 was predominantly found in the cytoplasmic staining of all CCA tissues. ACSL4, SLC7A11, and CHAC1 expression was also detected in normal bile ducts; however, their expression levels were found to be low. ACSL4 expression was low in 48.9% (67/137) of the cases, while high expression was observed in 51.1% (70/137) of the cases. SLC7A11 expression was low in 50.4% (69/137) of the cases, while a high expression was observed in 49.6% (68/137) of the cases. CHAC1 expression was low in 38.7% (53/137) of the cases, while high expression was observed in 61.3% (84/137) of the cases, as shown in Figure 1.

### 3.2. Association of ACLS4, SLC7A11, and CHAC1 Protein Levels with Clinicopathological Data and Laboratory Results

High ACSL4 expression levels showed a trend towards an association with the extrahepatic (*p* = 0.05) and mixed tumor growth (*p* = 0.06) types of CCA. Furthermore, patients with CCA who survived for 2 years with high ACSL4 expression were associated with the extrahepatic type (*p* = 0.04), the tumor growth type (*p* = 0.03), and alanine transferase (ALT) levels above the normal range (≥33 U/L; *p* = 0.05), as shown in Table 1. There was a significant association between high SLC7A11 expression and the tumor growth type (*p* = 0.01; Appendix A). In addition, there was a trend towards an association between high CHAC1 expression and elevated levels of direct bilirubin (*p* = 0.07) and between high CHAC1 expression and higher levels of ALT (*p* = 0.06) in patients with CCA who survived for 2 years, as shown in Table 1.

The association between the OS of patients with CCA and the expression levels of ACLS4, SLC7A11, and CHAC1 were analyzed using Kaplan–Meier survival curve analysis. No statistically significant association was observed between ACSL4 expression and the survival time of patients with CCA (*p* = 0.51; Figure 2A). The ACSL4 low-expression group had a median survival time of 15.8 months, while the ACSL4 high-expression group had a median survival time of 16.3 months. In addition, there was not a statistically significant association between the expression level of SLC7A11 and survival time (*p* = 0.10; Figure 2B). The median survival time of patients with low SLC7A11 expression was 13.0 months, while patients with high SLC7A11 expression had a median survival time of 17.0 months. Furthermore, there was not a statistically significant association between the expression level of CHAC1 and the survival time of patients with CCA (*p* = 0.69; Figure 2C). The median survival time of patients with low CHAC1 expression was 16.80 months, while patients with high CHAC1 expression had a median survival time of 13.2 months. 

Next, the association between ACLS4, SLC7A11, or CHAC1 expression and the survival time of patients with CCA was specifically assessed within a 2-year survival analysis period. There was no significant association between low or high ACLS4 expression and survival time, with a median survival time of 10.5 and 9.0 months, respectively (*p* = 0.30; Figure 2D). There was also no significant association between the SLC7A11 expression level and survival time (*p* = 0.76; Figure 2E). The median survival time of patients with low SLC7A11 expression was 10.3 months, while patients with high SLC7A11 expression had a median survival time of 9.2 months. However, high CHAC1 expression was significantly associated with a shorter survival time compared with low CHAC1 expression (*p* = 0.03; Figure 2F). The median survival time of patients with low CHAC1 expression was 12.8 months, while patients with high CHAC1 expression had a median survival time of 9.2 months.

### 3.3. Clinicopathological Variables Potentially Associated with the Survival Rate of Patients with CCA

In the univariate analysis, high SLC7A11 expression demonstrated a trend towards an association with patient survival rates [hazard ratio (HR) = 0.75; *p* = 0.10]. The univariate results showed that factors including cell type, lymph node metastasis, distant metastasis, and TMN stage were associated with the survival rate; as such, they were included in the multivariate analysis. The multivariate analysis results showed that TMN stage III–IV was significantly correlated with a shorter survival time than TMN stage I–II [HR = 2.39; 95% confidence interval (CI), 1.36–4.18; *p* < 0.01]. However, SLC7A11 was not associated with the survival rate of patients with CCA, as shown in Appendix A. 

The data were filtered and focused on patients with CCA at 2 years of survival. The univariate analysis revealed that sex (male), papillary type, lymph node metastasis, distant metastasis, TMN stage (III–IV), and high CHAC1 expression were associated with the survival rate; as such, they were included in the multivariate analysis. The multivariate results showed that high CHAC1 expression was significantly correlated with a shorter survival time when compared with the low CHAC1 expression group (HR = 1.98; 95% CI, 1.21–3.24; *p* < 0.01). In addition, papillary type (HR = 0.57; 95% CI, 0.36–0.90; *p* = 0.01) and distant metastasis (HR = 2.26; 95% CI, 0.91–2.73; *p* = 0.02) showed increased patient survival rate hazard ratios, as shown in Table 2.

### 3.4. ACLS4 and SLC7A11 Levels in CCA Sera

The levels of secreted ACSL4 and SLC7A11 proteins were measured in the sera obtained from two CCA patient groups: the early and advanced CCA groups. The advanced CCA group exhibited a significantly elevated level of ACSL4 compared with the early CCA group. The mean ACSL4 values in the early and advanced CCA groups were 0.70 and 0.97, respectively (Figure 3A). In addition, ACSL4 was capable of distinguishing advanced CCA from early CCA [area under the curve (AUC) = 0.803; *p* < 0.01; Figure 3B]. The ACSL4 level predicting the severity of CCA was categorized as high or low ACSL4 expression by using a cut-off value of 0.74, with a Youden index of 0.50, a sensitivity of 96.7%, and a specificity of 53.3% as shown in Table 3.

SLC7A11 also exhibited the ability to differentiate the early and advanced CCA groups. The mean SLC7A11 values in the early and advanced CCA groups were 0.71 and 0.95, respectively (Figure 3D). The AUC result also demonstrated that SLC7A11 could classify early and advanced CCA (AUC = 0.789; *p* < 0.01; Figure 3E). The SLC7A11 level predicting the severity of CCA was categorized as high or low SLC7A11 expression by using a cut-off value of 0.90, with a Youden index of 0.54, a sensitivity of 67.7%, and a specificity of 86.6%, as shown in Table 3.

### 3.5. Association of Clinicopathological Variables with ACSL4 and SLC7A11 Levels in CCA Sera 

There was a significant association between elevated levels of both ACSL4 and SLC7A11 in sera with advanced TMN stage (stage III-IV) and high bilirubin levels (*p* < 0.01). Elevated ACSL4 levels were also markedly associated with increased levels of aspartate aminotransferase (AST), ALT, and CA19-9 (*p* = 0.05, *p* = 0.05, and *p* = 0.02, respectively). There was a significant association between high SLC7A11 levels and a lower albumin level (*p* < 0.01), as shown in Table 4. In addition, a high ACSL4 level was significantly associated with a shorter survival time (*p* < 0.01; Figure 3C). The ACSL4 low-expression group had a median survival time of 42.97 months, while the ACSL4 high-expression group had a median survival time of 6.20 months. There was also a significant association between high SLC7A11 levels and survival time (*p* < 0.01; Figure 3F). Patients with low SLC7A11 levels had a median survival time of 24.10 months, while patients with high SLC7A11 levels had a median survival time of only 4.17 months.

## 4. Discussion

Ferroptosis is a type of controlled cell death characterized by the accumulation of ROS, through the formation of iron and lipid peroxides. Numerous mechanisms are associated with the ferroptosis process, such as iron metabolism, oxidation mechanisms (from lipid peroxidation), and antioxidant mechanisms. Several ferroptosis biomarkers have been reported, including transferrin receptor (TFRC), ACSL4, SLC7A11, and CHAC1 [24].

In the present study, three ferroptosis-related proteins, ACSL4, SLC7A11, and CHAC1, were selected for investigation to verify their potential as ferroptosis biomarkers. The ACSL4, SLC7A11, and CHAC1 expression levels were assessed in CCA tissues and sera. Immunohistochemical analysis was performed to evaluate the ACLS4, SLC7A11, and CHAC1 protein expression levels in CCA tissues. ACSL4 has a vital role in catalyzing long-chain fatty acid activation and lipid metabolism. ACSL4 is mainly localized to the cytoplasm, and a previous study reported that ACSL4 was not found in peroxisomes [25]. ACSL4 expression has been associated with a number of biological activities, including the inflammatory response, immune response, steroidogenesis, and cell death [26]. In addition, ACSL4 could play a controversial role in cancer. ACSL4 can promote tumor progression in various cancer types. It has been documented that low ACSL4 levels are correlated with a good prognosis in breast cancer [27]. In addition, a higher ACSL4 level was reported to be associated with a poorer prognosis in colorectal cancer and was correlated with shorter survival and DFS in HCC [28,29]. The results of the present study indicated that high ACSL4 levels were significantly associated with the extrahepatic type, the tumor growth type, and high ALT levels in CCA. These results are concordant with those of Toshida et al. [30], who also demonstrated that high ACSL4 expression was correlated with histological liver fibrosis. The upregulation of ACSL4 is associated with an advanced stage of prostate cancer and is also correlated with tumor invasion and migration in prostate cancer cells [31]. However, to the best of our knowledge, no study has reported a direct correlation between ACSL4 and ALT levels as a liver inflammation biomarker. A previous study reported a strong association between ACSL4 expression and fatty liver disease as a risk factor leading to liver cancer. Duan et al. [32] showed that ACSL4 knockout could suppress hepatic triglyceride levels and was correlated with decreasing serum ALT and AST levels in mice fed a high-fructose and high-fat diet to induce non-alcoholic steatohepatitis (NASH). Therefore, ACSL4 knockout may protect mice from developing NASH, which can lead to liver cancer.

SLC7A11 has been reported as a key regulator of ferroptosis. The essential role of SLC7A11 is in importing cystine or glutamate/cystine transport, which controls the homeostasis of ROS in cells [33]. Previous studies have shown that SLC7A11 might be a therapeutic target of drugs, such as erastin, in which the inhibition of SLC7A11 leads to ferroptotic cell death through ROS accumulation [34,35]. The role of SLC7A11 expression has been reported in several cancers. SLC7A11 expression has also been associated with severity and radiosensitivity in tissues of esophageal squamous cell carcinoma (ESCC) patients. There are results indicating that SLC7A11 overexpression is strongly associated with lymph node metastasis, short survival times, and poor treatment response [36]. Zhang et al. [33] reported an association between high SLC7A11 expression and advanced pathological differentiation in HCC tissues. Moreover, high SLC7A11 expression was associated with a shorter survival time, as compared with low SLC7A11 expression in HCC. In addition, SLC7A11 expression was reported to have a positive correlation with clinical data including lymph node metastasis status and disease recurrence status in tissues of colorectal cancer patients [37]. In concordance with our study, it was demonstrated that high SLC7A11 expression was significantly associated with the tumor growth type in CCA. The relationship between SLC7A11 expression and the OS of patients with CCA was also examined in the present study. The results indicated that there was no significant association between SLC7A11 expression and the survival rate of patients with CCA. A possible explanation for this may be that SLC7A11 expression in CCA serves more as a therapeutic target marker than as a prognostic biomarker.

CHAC1, an enzyme associated with the activity of γ-glutamylcyclotransferase, plays a crucial role in promoting the ferroptosis of tumor cells by degrading intracellular GSH. CHAC1 participates in the γ-glutamyl cycle, digesting GSH into 5-oxoproline and Cys-Gly dipeptide and consequently reducing GSH levels in cells. This reduction in GSH levels promotes both apoptosis and ferroptosis [16]. A recent study showed that the upregulation of CHAC1 was positively correlated with ALT and AST levels in mice with acute liver injury [38]. Similarly, the results of the present study confirmed that there was a trend towards an association of increased CHAC1 expression with elevated levels of ALT, as well as with higher levels of direct bilirubin, in the sera of patients with CCA who survived for 2 years. Moreover, the results of the survival analysis showed a significant association between high CHAC1 expression levels and a shorter survival time in patients with CCA with 2 years of survival, compared with low CHAC1 expression. This result is in concordance with a study by Mehta et al. [39], in which it was revealed that high CHAC1 expression was correlated with a poorer prognosis in breast cancer. Similarly, CHAC1 overexpression in breast cancer tissues is associated with lymph node metastasis and increased cell proliferation, indicating poor prognosis in breast cancer [40].

Several studies have reported correlations between survival rates and ACSL4, SLC7A11, and CHAC1 expression. In the present study, the results of the univariate analysis revealed that high SLC7A11 expression was associated with an increased HR of lower survival rates (HR = 0.75; *p* =0.10). In the univariate analysis focusing on patients with CCA with 2 years of survival, high CHAC1 levels were significantly associated with an increased HR, resulting in lower a survival rate (HR = 1.66; *p* < 0.01). The results of the multivariate analysis revealed a significant correlation between high CHAC1 levels and a shorter survival time when compared with the low CHAC1 group (HR = 1.98; *p* < 0.01). Similarly, in a study by Li et al. [41], the results of a multivariate Cox regression analysis revealed that CHAC1 was associated with OS in kidney renal clear-cell carcinoma. CHAC1 may therefore be used as a prognostic marker for medical management or as a therapeutic target.

Using serum to determine protein levels is a non-invasive technique, and serum samples are easier to collect from patients than a biopsy. However, the correlation between the expression of ACSL4 and SLC7A11 in sera and cancer patients’ prognoses is still less explored in publications and remains controversial in several diseases. A previous study indicated that a combined high ACSL4 level and low-dose 256-slice spiral CT may be a potential biomarker for the screening of early lung cancer, and high ACSL4 indicates a better prognosis than low ACSL4 levels [19]. This result was opposed by Hu et al. [42], who demonstrated that elevated ACSL4 levels were significantly associated with ST-segment elevation myocardial infarction when compared with a group with normal ACSL4 levels. Furthermore, high ACSL4 levels were correlated with 1-year major adverse cardiovascular events in patients of the poorer prognosis group compared with the favorable group. The results of the present study revealed that the advanced CCA group exhibited significantly elevated levels of ACSL4 and SLC7A11 in CCA sera compared with the early CCA group. ACSL4 and SLC7A11 were therefore capable of distinguishing advanced CCA from early CCA. Elevated ACSL4 and SLC7A11 levels in CCA sera were also associated with a higher staging, which indicated a poorer prognosis in patients with CCA. Furthermore, high SLC7A11 levels in CCA sera were associated with high total bilirubin and low albumin levels, indicating a poorer prognosis in patients with CCA. This result is in agreement with the results of a study by Zhan et al. [20], in which it was shown that a high SLC7A11 level was associated with advanced stage and low albumin levels in HCC. SLC7A11 levels were also shown to be a complementary marker for alpha fetoprotein (AFP) levels in the diagnosis of HCC from liver cirrhosis. According to the findings of the present study, both ACSL4 and SLC7A11 levels may serve as complementary markers in patients with CCA with low CEA and CA19-9 levels. Nevertheless, the present study was limited by the small sample size of CCA sera, and further validation should be conducted to confirm the ACLS4 and SLC7A11 levels in CCA sera.

## 5. Conclusions

The results of the present study demonstrated a significant association between high ACSL4 levels and the extrahepatic and tumor growth types of CCA, as well as elevated levels of ALT. Significant associations were also found between elevated SLC7A11 levels and the tumor growth type. A high CHAC1 level was significantly associated with a short survival time in patients with CCA. In addition, high levels of ACSL4 and SLC7A11 in CCA sera were significantly associated with advanced tumor staging, poorer prognosis parameters, and a short survival time. Taken together, ACSL4, SLC7A11, and CHAC1 may be used as a valuable biomarker for predicting prognosis when CCA tissues are assessed. In addition, ACSL4 and SLC7A11 are potential complementary biomarkers for predicting prognosis when CCA sera are assessed. This study provides good basic information for further study. In future research, the sample size should be expanded to investigate the role of ACSL4 and SLC7A11 and their inhibitors as a targeted therapy for improved therapeutic planning and management in patients with CCA. 

## Figures and Tables

**Figure 1 biomedicines-12-02091-f001:**
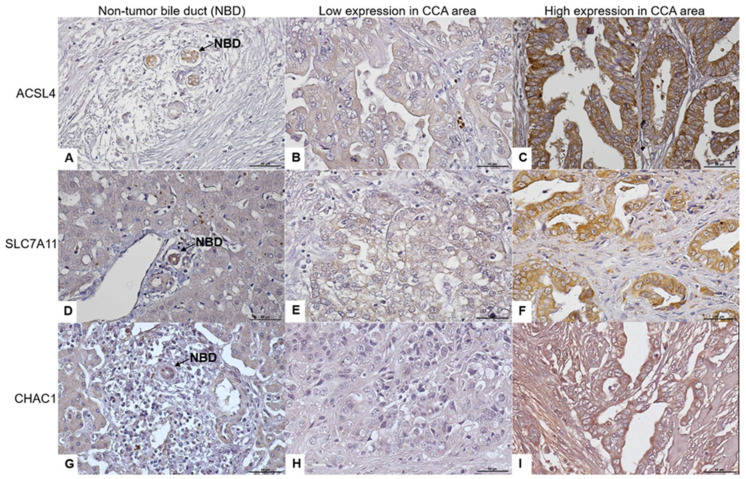
Immunohistochemical staining for ACSL4, SLC7A11, and CHAC1 protein expression in human CCA tissues. (**A**–**C**) show the ACSL4 expression in a non-tumor bile duct and CCA area. (**D**–**F**) show the SLC7A11 expression in a non-tumor bile duct and CCA area. (**G**–**I**) show the CHAC1 expression in a non-tumor duct and CCA area. NBD: non-tumor bile duct; scale bar: 50 μm; magnification: ×40.

**Figure 2 biomedicines-12-02091-f002:**
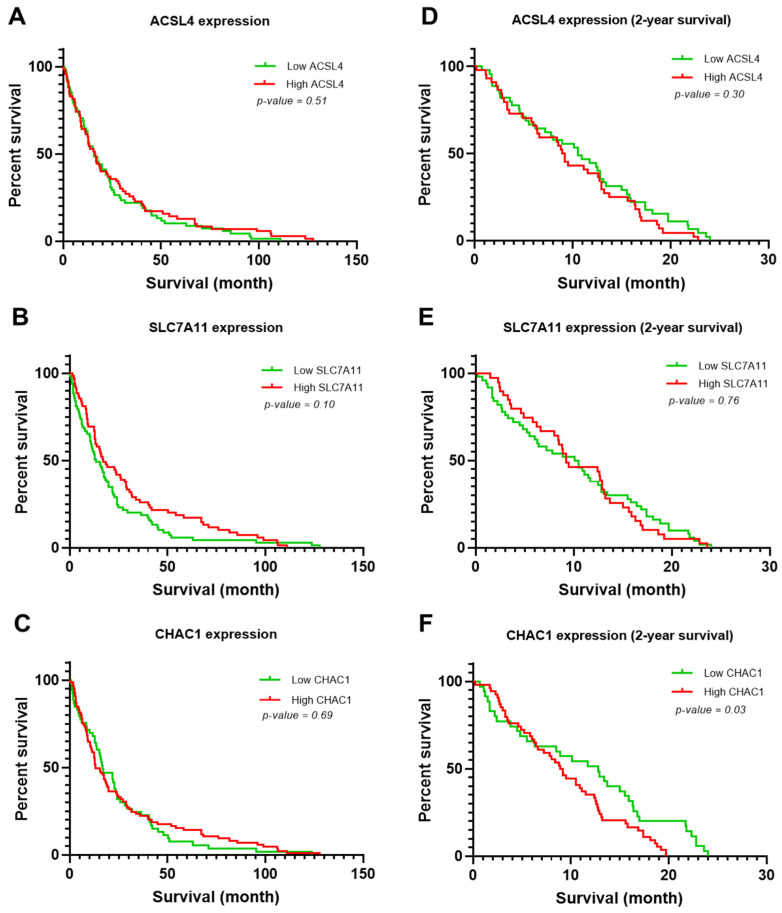
Overall survival analysis of ACSL4, SLC7A11, and CHAC1 expression in all CCA patients and 2-year survival analysis. (**A**) Survival analysis of ACSL4 expression in CCA patients. (**B**) Survival analysis of SLC7A11 expression in CCA patients. (**C**) Survival analysis of CHAC1 expression in all CCA patients. (**D**) Survival analysis of ACSL4 expression in CCA 2-year survival. (**E**) Survival analysis of SLC7A11 expression in CCA 2-year survival. (**F**) Survival analysis of CHAC1 expression in CCA 2-year survival. All analyzed data used the median of the H-score as the cut-off.

**Figure 3 biomedicines-12-02091-f003:**
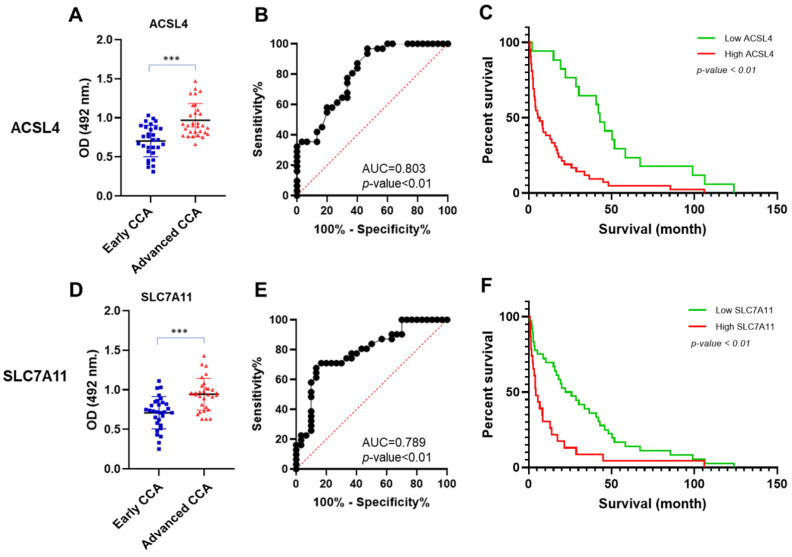
The level of ACSL4 and SLC7A11 expression and overall survival analysis in advanced CCA compared to early CCA in sera. There was a significant correlation found between the elevated levels of ACSL4 and SLC7A11 in the advanced stage group and the short survival of CCA patients. (**A**) ACSL4 expression was significantly higher in advanced CCA sera than in early CCA sera. (**B**) The AUC in different groups based on ACLS4 expression. (**C**) Survival analysis of ACSL4 expression in CCA sera patients. (**D**) SLC7A11 expression was significantly higher in advanced CCA sera than in early CCA sera. (**E**) The AUC in different groups based on SLC7A11 expression. (**F**) Survival analysis of SLC7A11 expression in CCA sera patients. All analyzed data used a cut-off calculated by the YI.

**Table 1 biomedicines-12-02091-t001:** Correlation between ACSL4, SLC7A11, and CHAC1 expression and clinicopathological data of CCA patients (2-year survival analysis).

Characteristics	ACSL4 Expression			SLC7A11 Expression			CHAC1 Expression		
	Low n, (%)	High n, (%)	*p*-Value	Low n, (%)	High n, (%)	*p*-Value	Low n, (%)	High n, (%)	*p*-Value
Gender			0.45			0.27			0.66
Female	15 (45.5%)	18 (54.5%)		21 (63.6%)	12 (36.4%)		12 (36.4%)	21 (63.6%)	
Male	30 (53.6%)	26 (46.4%)		29 (51.8%)	27 (48.2%)		23 (41.1%)	33 (58.9%)	
Age (median, years)			0.75			0.62			0.96
<60	21 (48.8%)	22 (51.2%)		23 (53.5%)	20 (46.5%)		17 (39.5%)	26 (60.5%)	
≥60	24 (52.2%)	22 (47.8%)		27 (58.7%)	19 (41.3%)		18 (39.1%)	28 (60.9%)	
Location of tumor			**0.04**			0.24			0.40
Intrahepatic	38 (56.7%)	29 (43.3%)		40 (59.7%)	27 (40.3%)		28 (41.8%)	39 (58.2%)	
Extrahepatic	7 (31.8%)	15 (68.2%)		10 (45.5%)	12 (54.5%)		7 (31.8%)	15 (68.2%)	
Tumor growth type			**0.03**			0.12			0.38
Intraductal type	11 (73.3%)	4 (26.7%)		9 (60.0%)	6 (40.0%)		6 (40.0%)	9 (60.0%)	
Mass-forming type	15 (57.7%)	11 (42.3%)		19 (73.1%)	7 (26.9%)		14 (53.8%)	12 (46.2%)	
Mixed type	14 (36.8%)	24 (63.2%)		18 (47.4%)	20 (52.6%)		14 (36.8%)	24 (63.2%)	
Cell type			0.45			0.30			0.51
Papillary	23 (54.8%)	19 (45.2%)		26 (61.9%)	16 (38.1%)		15 (35.7%)	27 (64.3%)	
Non-papillary	22 (46.8%)	25 (53.2%)		24 (51.1%)	23 (48.9%)		20 (42.6%)	27 (57.4%)	
Lymph node metastasis			0.93			0.17			0.85
Yes	29 (50.9%	28 (49.1%)		29 (50.9%	28 (49.1%)		22 (38.6%	35 (61.4%)	
No	16 (50.0%)	16 (50.0%)		29 (50.9%	28 (49.1%)		13 (40.6%)	19 (59.4%)	
Distant metastasis			0.52			0.50			0.15
Yes	4 (40.0%)	6 (60.0%)		7 (70.0%)	3 (30.0%)		6 (60.0%)	4 (40.0%)	
No	41 (51.9%)	38 (48.1%)		43 (54.4%)	36 (45.6%)		29 (36.7%)	50 (63.3%)	
TMN stage			0.95			0.63			0.61
I–II	9 (50.0%)	9 (50.0%)		11 (61.1%)	7 (38.9%)		8 (44.4%)	10 (55.6%)	
III–IV	36 (50.7%)	35 (49.3%)		39 (54.9%)	32 (45.1%)		27 (38.0%)	44 (62.0%)	
OV infection			0.65			0.52			0.33
Yes	34 (49.3%)	35 (50.7%)		40 (58%)	29 (42%)		29 (42.0%)	40 (58.0%)	
No	34 (49.3%)	35 (50.7%)		10 (50%)	10 (50%)		6 (30.0%)	14 (70.0%)	
Total protein (g/dL)			0.26			0.11			0.90
<8.7	35 (47.3%)	39 (52.7%)		38 (51.4%)	36 (48.6%)		30 (40.5%)	44 (59.5%)	
≥8.7	5 (71.4%)	2 (28.6%)		6 (85.7%)	1 (14.3%)		3 (42.9%)	4 (57.1%)	
Globulin (g/dL)			0.82			0.42			0.40
<3.4	16 (48.5%)	17 (51.5%)		16 (48.5%)	17 (51.5%)		15 (45.5%)	18 (54.5%)	
≥3.4	24 (51.1%)	23 (48.9%)		27 (57.4%)	20 (42.6%)		17 (36.2%)	30 (63.8%)	
Total bilirubin (mg/dL)			0.48			0.41			0.18
<1.2	31 (51.7%)	29 (48.3%)		31 (51.7%)	29 (48.3%)		27 (45.0%)	33 (55.0%)	
≥1.2	9 (42.9%)	12 (57.1%)		13 (61.9%)	8 (38.1%)		6 (28.6%)	15 (71.4%)	
Direct bilirubin (mg/dL)			0.23			0.92			0.07
<0.5	26 (54.2%)	22 (45.8%)		26 (54.2%)	22 (45.8%)		23 (47.9%)	25 (52.1%)	
≥0.5	13 (40.6%)	19 (59.4%)		17 (53.1%)	15 (46.9%)		9 (28.1%)	23 (71.9%)	
ALT (U/L)			0.05			0.46			0.06
<33	20 (62.5%)	12 (37.5%)		19 (59.4%)	13 (40.6%)		17 (53.1%)	15 (46.9%)	
≥33	20 (40.8%)	29 (59.2%)		25 (51.0%)	24 (49.0%)		16 (32.7%)	33 (67.3%)	
AST (U/L)			0.22			0.97			0.10
<40	19 (57.6%)	14 (42.4%)		18 (54.5%)	15 (45.5%)		17 (51.5%)	16 (48.5%)	
≥40	21 (43.8%)	27 (56.3%)		26 (54.2%)	22 (45.8%)		16 (33.3%)	32 (66.7%)	
ALP (U/L)			0.77			0.97			0.88
<129	16 (53.3%)	14 (46.7%)		17 (56.7%)	13 (43.3%)		13 (43.3%)	17 (56.7%)	
≥129	24 (50.0%)	24 (50.0%)		27 (56.3%)	21 (43.8%)		20 (41.7%)	28 (58.3%)	

Statistically significant *p*-values are shown in bold.

**Table 2 biomedicines-12-02091-t002:** The univariate and multivariate analysis of clinicopathological data (2-year survival).

Variable	Overall Survival							
	Univariate				Multivariate			
	HR	95% CI		*p*-Value	HR	95% CI		*p*-Value
Age (≥60)	0.81	0.53	1.23	0.33				
Gender (Male)	0.76	0.49	1.18	0.22				
Location of tumor (Intrahepatic)	0.96	0.59	1.57	0.90				
Cell types (Papillary)	0.52	0.34	0.82	**<0.01**	0.57	0.36	0.90	**0.01**
Tumor growth type (Mixed type)	1.27	0.68	2.34	0.44				
Lymph node metastasis (Yes)	1.42	0.91	2.22	0.11				
Distant metastasis (Yes)	2.11	1.07	4.15	**0.03**	2.26	1.10	4.64	**0.02**
TMN stage (III–IV)	1.81	1.06	3.09	**0.03**	1.57	0.91	2.73	0.10
OV infection (Positive)	0.83	0.50	1.38	0.48				
ACSL4 (High)	1.24	0.81	1.90	0.30				
SLC7A11 (High)	1.06	0.69	1.62	0.77				
CHAC1 (High)	1.66	1.04	2.65	**0.03**	1.98	1.21	3.24	**<0.01**

Statistically significant *p*-values are shown in bold.

**Table 3 biomedicines-12-02091-t003:** Predictive values of ACSL4 and SLC7A11 serum levels for predictive prognosis in CCA, based on the optimal cut-off derived from ROC analysis and Youden index calculations.

Group Comparison	Protein	AUC (95% CI)	Cut-Off (OD)	Youden Index	Sensitivity (%)	Specificity (%)	LR	*p*-Value
Early CCA vs. Advanced CCA	ACSL4	0.80 (0.69–0.91)	0.74	0.50	96.7	53.3	2.07	**<0.01**
	SLC7A11	0.70 (0.67–0.90)	0.90	0.54	67.7	86.6	5.08	**<0.01**

Statistically significant *p*-values are shown in bold.

**Table 4 biomedicines-12-02091-t004:** Correlation between ACSL4 and SLC7A11 expression and clinicopathological data in CCA sera.

Characteristics	ACSL4 Expression			SLC7A11 Expression		
	Low n, (%)	High n, (%)	*p*-Value	Low n, (%)	High n, (%)	*p*-Value
Gender			0.38			0.12
Female	6 (22.2%)	21 (77.8%)		13 (48.1%)	14 (51.9%)	
Male	11 (32.4%)	23 (67.6%)		23 (67.6%)	11 (32.4%)	
Age (median, years)			0.57			0.34
<60	8 (47.1%)	9 (52.9%)		13 (76.5%)	4 (23.5%)	
≥60	8 (57.1%)	6 (42.9%)		13 (92.9%)	1 (7.1%)	
TMN stage			**<0.01**			**<0.01**
I–II	16 (53.3%)	14 (46.7%)		26 (86.7%)	4 (13.3%)	
III–IV	1 (3.2%)	30 (96.8%)		10 (32.3%)	21 (67.7%)	
Albumin (g/dL)			0.11			**<0.01**
<3.5	4 (16.0%)	21 (84.0%)		8 (32.0%)	17 (68.0%)	
≥3.5	11 (34.4%)	21 (65.6%)		24 (75.0%)	8 (25.0%)	
Total bilirubin (mg/dL)			**<0.01**			**<0.01**
<1.2	15 (57.7%)	11 (42.3%)		22 (84.6%)	4 (15.4%)	
≥1.2	1 (3.1%)	31 (96.9%)		11 (34.4%)	21 (65.6%)	
Direct bilirubin (mg/dL)			**<0.01**			**<0.01**
<0.5	14 (60.9%)	9 (39.1%)		19 (82.6%)	4 (17.4%)	
≥0.5	2 (5.7%)	33 (94.3%)		14 (40.0%)	21 (60.0%)	
ALT (U/L)			0.53			0.66
<33	6 (33.3%)	12 (66.7%)		11 (61.1%)	7 (38.9%)	
≥33	10 (25.0%)	30 (75.0%)		22 (55.0%)	18 (45.0%)	
AST (U/L)			0.05			0.09
<40	9 (42.9%)	12 (57.1%)		15 (71.4%)	6 (28.6%)	
≥40	7 (18.9%)	30 (81.1%)		18 (48.6%)	19 (515.4%)	
ALP (U/L)			0.05			0.17
<129	8 (47.1%)	9 (52.9%)		12 (70.6%)	5 (29.4%)	
≥129	8 (19.5%)	33 (80.5%)		21 (51.2%)	20 (48.8%)	
CA19-9 (U/mL)			**0.02**			0.13
<37	10 (58.8%)	7 (41.2%)		13 (76.5%)	4 (23.5%)	
≥37	4 (21.1%)	15 (78.9%)		10 (52.6%)	9 (47.4%)	
CEA (ng/mL)			0.93			0.46
<2.5	5 (38.5%)	8 (61.5%)		8 (61.5%)	5 (38.5%)	
≥2.5	8 (40.0%)	12 (60.0%)		15 (75.0%)	5 (25.0%)	

Statistically significant *p*-values are shown in bold.

## Data Availability

The data that support the findings of this study are available from the corresponding author(s) upon reasonable request.

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
