# Peer review of "Prognostic Values of Ferroptosis-Related Proteins ACSL4, SLC7A11, and CHAC1 in Cholangiocarcinoma"

_biomedicines, 2024, doi:10.3390/biomedicines12092091_

Round 1

Reviewer 1 Report

Comments and Suggestions for Authors

The manuscript presents an investigation into the prognostic value of ferroptosis-related proteins ACSL4, SLC7A11, and CHAC1 in cholangiocarcinoma (CCA). While the study attempts to address a significant gap in CCA biomarker research, it suffers from critical flaws that necessitate major revision. The study's design is fundamentally weakened by the absence of appropriate control groups, which limits the ability to assert the specificity of these proteins as prognostic markers for CCA. Additionally, the methodological details, particularly concerning antibody validation in immunohistochemistry, are insufficiently described, raising concerns about the reliability of the findings. The statistical analyses, while providing some significant results, are underpowered due to the small sample size, particularly in the serum analysis. This increases the likelihood of type II errors, and the broad confidence intervals suggest that the findings may be overstated. The discussion lacks depth, particularly in exploring the mechanistic pathways that could explain the observed associations. Furthermore, the novelty of the work is limited, as it primarily reiterates known information without offering substantial new insights specific to CCA. A comprehensive revision addressing these issues is essential to enhance the study's validity and impact on the field.

1. The absence of a control group is a critical flaw that undermines the study's validity. Without comparisons to benign or normal tissues, the specificity of the biomarkers remains unproven.

2. The claim that ACSL4 is associated with poorer prognosis due to elevated ALT levels is speculative. The authors fail to provide sufficient evidence or relevant citations linking ACSL4 expression directly to ALT elevation in CCA. While the study attempts to draw parallels with liver inflammation biomarkers, the rationale remains weak without supporting literature or mechanistic insights. The link between ACSL4 and ALT levels, although noted, does not establish causality. The connection is inferred rather than demonstrated, making the conclusion premature. Further, the authors should investigate the potential confounding variables that might influence ALT levels, such as co-existing liver conditions or the presence of metastasis. The lack of exploration into these factors detracts from the study's depth. Incorporating a more detailed analysis and discussion regarding these associations would greatly enhance the study’s credibility. The superficial treatment of these associations diminishes the overall impact of the study's findings.

3. The sample size of the serum analysis is insufficient for meaningful conclusions. With only 61 samples, the study is likely underpowered, increasing the risk of type II errors and limiting the generalizability of the results. Larger cohorts are necessary for robust statistical analysis.

4. The methodology description is vague, particularly concerning the antibody validation process. The authors should include details on how they confirmed the specificity of the antibodies used for ACSL4, SLC7A11, and CHAC1 to ensure the reliability of their findings.

5. The study's contribution to the field is minimal. The research reiterates known associations between ferroptosis-related proteins and cancer prognosis without providing significant new insights specific to CCA. The novelty of this work is questionable.

6. The statistical significance reported for CHAC1's impact on survival time (HR = 1.98, p < 0.01) is undercut by broad confidence intervals. This suggests inadequate control of variables and potential overestimation of the marker's prognostic value, casting doubt on the findings.

7. The use of immunohistochemistry alone to assess protein expression levels is insufficient. The study should have included additional techniques, such as Western blotting or qPCR, to validate the findings and strengthen the evidence for the prognostic value of ACSL4, SLC7A11, and CHAC1. This omission is a significant oversight.

8. The interpretation of the results lacks depth, particularly concerning the non-significant associations between ACSL4, SLC7A11, and patient survival. The authors fail to discuss the implications of these findings adequately, weakening the overall argument for the prognostic value of these markers in CCA. Without a thorough exploration of these results, the study's conclusions are less convincing and may not hold up to further scrutiny.

Comments on the Quality of English Language

Moderate editing of English language required.

Author Response

Reviewer 1 comments

Point 1: The absence of a control group is a critical flaw that undermines the study's validity. Without comparisons to benign or normal tissues, the specificity of the biomarkers remains unproven.

Author’s response:

We appreciate and agree to the Reviewer’s suggestion. Regretfully, because we are unable to obtain tissues from healthy individuals. Due to tissue microarray being used in this study, only a limited number of adjacent normal areas (non-tumor bile duct; NBD in the figure) were observed.  Low expression level of ACSL4, SLC7A11, and CHAC1 were seen in that area. We have added the figure that represent the non-tumor bile duct or adjacent normal tissues in each protein as shown in Figures 1A, 1D, and 1G in the result section of the main manuscript.

Figure 1 Immunohistochemical staining for ACSL4, SLC7A11, and CHAC1 protein expression in human CCA tissues. (A-C) Shown the ACSL4 expression in non-tumor bile duct and CCA area. (D-F) Shown the SLC7A11 expression in non-tumor bile duct and CCA area. (G-I) Shown the CHAC1 expression in non-tumor duct and CCA area. NBD: non-tumor bile duct; scale bar: 50 μm; magnification ×40.

Figure 1 Immunohistochemical staining for ACSL4, SLC7A11, and CHAC1 protein expression in human CCA tissues. (A-C) Shown the ACSL4 expression in non-tumor bile duct and CCA area. (D-F) Shown the SLC7A11 expression in non-tumor bile duct and CCA area. (G-I) Shown the CHAC1 expression in non-tumor duct and CCA area. NBD: non-tumor bile duct; scale bar: 50 μm; magnification ×40.

Point 2: The claim that ACSL4 is associated with poorer prognosis due to elevated ALT levels is speculative. The authors fail to provide sufficient evidence or relevant citations linking ACSL4 expression directly to ALT elevation in CCA. While the study attempts to draw parallels with liver inflammation biomarkers, the rationale remains weak without supporting literature or mechanistic insights. The link between ACSL4 and ALT levels, although noted, does not establish causality. The connection is inferred rather than demonstrated, making the conclusion premature. Further, the authors should investigate the potential confounding variables that might influence ALT levels, such as co-existing liver conditions or the presence of metastasis. The lack of exploration into these factors detracts from the study's depth. Incorporating a more detailed analysis and discussion regarding these associations would greatly enhance the study’s credibility. The superficial treatment of these associations diminishes the overall impact of the study's findings.

Author’s response: We appreciate the Reviewer's suggestion. This study makes an interesting case for the potential association between ACSL4 and prognosis, particularly with ALT levels. While the direct evidence linking ACSL4 expression to ALT elevation in CCA is still emerging, our approach highlights an important area for future research.  ACSL4 expression and ALT levels may be correlated in term of co-incident expression case which still has no answer in biological relationship. One explanation is ACSL4 expression and ALT levels might be correlated in form of inflammation biomarkers in the relationship between liver cell and bile duce cell led to positive correlation outcome in our study. However, our core values indicating that ACSL4 is additional marker to support the results of elevated ALT levels to predicting prognosis of CCA patients. We have identified a possible connection between ACSL4 and ALT levels, which, although not yet fully demonstrated, provides a promising direction for further investigation. Recognizing the potential impact of confounding variables, such as co-existing liver conditions or metastasis, is a valuable insight that could enhance future research. However, the investigation of the relationship between ACSL4 and ALT and their mechanisms should be done in a future study.

Point 3: The sample size of the serum analysis is insufficient for meaningful conclusions. With only 61 samples, the study is likely underpowered, increasing the risk of type II errors and limiting the generalizability of the results. Larger cohorts are necessary for robust statistical analysis.

Author’s response:

We acknowledge the Reviewer’s concern. We calculate the type II error probability (β) and power of the test (1-β) from the mean and S.D. of each group in each protein. We found that our result has a power of tests 0.99 for both ACSL4 and SLC7A11 determination. We are confident that our results can be used as a population representative for preliminary screening to predict disease prognosis. However, as the Reviewer’s suggestion, larger cohorts are required for further study.

 Point 4: The methodology description is vague, particularly concerning the antibody validation process. The authors should include details on how they confirmed the specificity of the antibodies used for ACSL4, SLC7A11, and CHAC1 to ensure the reliability of their findings.

Author’s response:

We appreciate the reviewer’s concern. Based on our literature review, we have selected the commercial antibody with the highest popularity for immunohistochemistry field research. Those three antibodies have been used to recheck the specificity of other past studies with other techniques as shown in Table 1R. Additionally, each round of the immunohistochemical procedure used a negative control staining, and as Figure 1R illustrates, no staining was seen in that control. We believe that our antibodies targeting ACSL4, SLC7A11, and CHAC1 show specificity towards our experimental outcome.

Table 1R Antibodies review

Antibodies

Cat. number

Method

references

ACSL4

Abcam

#ab155282

-   Western blotting

-   Immunofluorescence

[1]

-   Immunohistochemistry

-   Western blotting

-   Immunofluorescence

[2]

SLC7A11

Abcam

#ab37185

-   Immunohistochemistry

-   Small Interfering RNA Transfection

-   Western Blotting

[3]

-   Knockout and overexpression of SLC7A11 in HepG2 cells

-   Western blotting

[4]

-   Immunohistochemistry

-   Western blotting

[5]

CHAC1

Protein tech

#15207-1-AP

-   Immunohistochemistry

[6]

-   RNA interferences

-   Western blotting

[7]

Figure 1R Immunohistochemical staining with negative control for ACSL4, SLC7A11, and CHAC1 protein expression in human CCA tissues. scale bar: 50 μm; magnification ×40.

Figure 1R Immunohistochemical staining with negative control for ACSL4, SLC7A11, and CHAC1 protein expression in human CCA tissues. scale bar: 50 μm; magnification ×40.

Point 5: The study's contribution to the field is minimal. The research reiterates known associations between ferroptosis-related proteins and cancer prognosis without providing significant new insights specific to CCA. The novelty of this work is questionable.

Author’s response: We thank the Reviewer's suggestion. This study is a study of proteins involved in ferroptosis in cholangiocarcinoma, a common cancer in the northeast of Thailand and a major public health problem of the country. First, this study is an initial study of the protein in cholangiocarcinoma (CCA) to determine whether its expression is related to prognosis. To find indicators that can be used as targeting molecules for the future treatment of CCA. For example, previous study showed ACSL4 expression was significantly associate with sorafenib as a ferroptosis inducer. The result showed high ACSL4 expression in tissues associated with a good responsive in HCC patient who receive sorafenib [8]. In addition, SLC7A11 was also investigated in role therapeutic target of drugs, such as erastin, in which inhibition of SLC7A11 leads to ferroptotic cell death through ROS accumulation for eliminate cancer cells [9,10].

Interestingly, ACSL4 and SLC7A11 expression in cholangiocarcinoma by using serum as a liquid biopsy in was not reported in previous. We are the first report to found that both proteins ACSL4 and SLC7A11 could be used as good potential biomarkers for predicting cholangiocarcinoma prognosis and treatment planning. Furthermore, further research is required to investigate the mechanism and inhibition of the expression of these proteins for the management of CCA.

Point 6: The statistical significance reported for CHAC1's impact on survival time (HR = 1.98, p < 0.01) is undercut by broad confidence intervals. This suggests inadequate control of variables and potential overestimation of the marker's prognostic value, casting doubt on the findings.

Author’s response: We appreciate the Reviewer's suggestion. Although the broad confidence interval was observed in this study which may be defected by an uncontrol variable. Nevertheless, our study demonstrated that higher CHAC1 protein expression was significantly correlated with a slight increase in HR, resulting in a 1.98-fold reduction in survival.

Point 7: The use of immunohistochemistry alone to assess protein expression levels is insufficient. The study should have included additional techniques, such as Western blotting or qPCR, to validate the findings and strengthen the evidence for the prognostic value of ACSL4, SLC7A11, and CHAC1. This omission is a significant oversight.

Author’s response: We appreciate and agree to the Reviewer’s concern. Unfortunately, due to the limitation of our laboratory for chemicals and equipment preparation. Further study on additional techniques as the Reviewer’s suggestion is required. However, we already review the mRNA expression through online database (GEPIA2 analysis) the result show that mRNA expression of ACSL4, SLC7A11, and CHAC1 was increased in CCA compared with the normal group. The results from the Kaplan-Meier survival analysis demonstrated that patients with CCA with high ACSL4 expression had a significantly lower OS and DFS compared with patients with low expression. It was also noted that elevated SLC7A11 levels were associated with a markedly reduced survival exclusively in patients with CCA as shown in Figure 2R. Our result indicating that the level on ACSL4, SLC7A11, and CHAC1 expression in CCA tissues was concordance with mRNA expression of three proteins via online database.

The mRNA expression level of ACSL4, SLC7A11, and CHAC1 in CCA compared to normal group via GEPIA2 analysis. And the survival analysis and disease-free survival of mRNA expression level of ACSL4, SLC7A11, and CHAC1 in CCA. The (A) ACSL4, (B) SLC7A11, and (C) CHAC1 mRNA expression level in CCA compared to normal group. (D) The overall survival and disease-free survival analysis of ACSL4, SLC7A11, and CHAC1 by Kaplan-Meier method with a log-rank test from the GEPIA2 database in CCA. Red color referred to cancer group, and green color referred to normal group.

Figure 2R The mRNA expression level of ACSL4, SLC7A11, and CHAC1 in CCA compared to normal group via GEPIA2 analysis. And the survival analysis and disease-free survival of mRNA expression level of ACSL4, SLC7A11, and CHAC1 in CCA. The (A) ACSL4, (B) SLC7A11, and (C) CHAC1 mRNA expression level in CCA compared to normal group. (D) The overall survival and disease-free survival analysis of ACSL4, SLC7A11, and CHAC1 by Kaplan-Meier method with a log-rank test from the GEPIA2 database in CCA. Red color referred to cancer group, and green color referred to normal group. 

Point 8: The interpretation of the results lacks depth, particularly concerning the non-significant associations between ACSL4, SLC7A11, and patient survival. The authors fail to discuss the implications of these findings adequately, weakening the overall argument for the prognostic value of these markers in CCA. Without a thorough exploration of these results, the study's conclusions are less convincing and may not hold up to further scrutiny.

Author’s response: We acknowledge the Reviewer’s comment. Our study provides valuable insights into the associations between ACSL4, SLC7A11, and patient survival in CCA tissues although not statistically significant but there is clinically significant. However, the elevated of ACSL4, and SLC7A11 expression in CCA tissues than adjacent normal area might be used as a potential additional marker for predict drug response and target therapy. Based on literature review, previous study showed ACSL4 expression was significantly associate with sorafenib as a ferroptosis inducer. The result showed high ACSL4 expression in tissues associated with a good responsive in HCC patient who receive sorafenib [8]. In addition, SLC7A11 was also investigated in role therapeutic target of drugs, such as erastin, in which inhibition of SLC7A11 leads to ferroptotic cell death through ROS accumulation for eliminate cancer cells [9,10]. From previous reported demonstrated that ACSL4 and SLC7A11 levels suitable for investigated in role of targeted therapy in further study. Moreover, ACSL4 and SLC7A11 expression in cholangiocarcinoma by using serum as a liquid biopsy in was not reported in previous. We are the first report to found that both proteins ACSL4 and SLC7A11 could be used as good potential biomarkers for predicting cholangiocarcinoma prognosis and treatment planning. Our approach opens up new avenues for further investigation and highlights the complexity of these biomarkers' roles in cancer prognosis. By presenting these findings, the study encourages future research to delve deeper into the potential prognostic value of ACSL4 and SLC7A11, laying the basis for more comprehensive analyses and discussions. This work serves as a basis for advancing our understanding of these markers in CCA.

Reviewer 2 Report

Comments and Suggestions for Authors

Comments to the Author

1.     The author should provide more background to the introduction part to strengthen the study.  For e.g., author should also focus on clinical relevance of CCA and why ferroptosis-related biomarkers are particularly significant for this type of cancer. This would help to better understand the study to readers who might not be familiar with ferroptosis or CCA.

2.     Author should add more detailed comparison study in the discussion part with existing literatures, particularly the studies that have been explored similar biomarkers in other cancers. This would help in highlighting the novelty of the study.

3.     Author should not only summarize the findings in the conclusion section but also suggest specific future research directions.

4.     Again read the whole manuscript to correct the grammatical mistakes. Improvement in the grammatical mistakes will enhance the readability and professionalism of the article.

5.     Why has author mentioned some digits in bold text in Table 1, 2, 3, 4? If this indicates some significance, then it should be clearly mentioned in the table legend.

6.     In line 361, “Zhang et al” was used, whereas “Mehta et al” was used in line 382. Correct them.

Comments on the Quality of English Language

Extensive editing of English language required.

Author Response

Response to Reviewer

Reviewer 2 comments

Point 1: The author should provide more background to the introduction part to strengthen the study.  For e.g., author should also focus on clinical relevance of CCA and why ferroptosis-related biomarkers are particularly significant for this type of cancer. This would help to better understand the study to readers who might not be familiar with ferroptosis or CCA.

Author’s response:

We appreciate and agree to the Reviewer’s suggestion. We have added the correlation and crucial roles of ferroptosis-related biomarkers and cancers in Background (line 79-88) as following:

(line 79-88)

The main role of ferroptosis is to eliminate abnormal cells, such as rapidly growing cells or cancer cells. For cancer cells in particular, ferroptosis is a double-edged sword. Cancer cells require higher levels of iron than normal cells, which can cause ferroptosis. On the other hand, cancer cells need to survive and avoid the ferroptosis process. Cancer cells attempt to increase antioxidant levels to reduce ROS and lipid oxidation levels, which are associated with oxidative stress. As a result, some ferroptosis-related proteins are upregulated, and cancer cells are not eliminated. Interestingly, the up-regulation of these ferroptosis-related proteins may be a weakness in cancer cells that can be targeted to provide biomarkers for therapeutic treatment [11,12].

Point 2: Author should add more detailed comparison study in the discussion part with existing literatures, particularly the studies that have been explored similar biomarkers in other cancers. This would help in highlighting the novelty of the study.

Author’s response:

We acknowledge the Reviewer’s suggestion. We have discused and rearrange some part of biomarkers in other cancers in Discussion part (line 371-383), (line 401-403) and we have added a sentence in (line 417-419) for clarity and to the point of pointing out that the expression of ACSL4 and SLC7A11 in CCA is interesting and has not been studied before as following:

(line 371-383)

The role of SLC7A11 expression has been reported in several cancers. SLC7A11 ex-pression has also been associated with severity and radiosensitivity in tissues of esophageal squamous cell carcinoma (ESCC) patients. There are results indicating that SLC7A11 overexpression is strongly associated with lymph node metastasis, short survival times, and poor treatment response [13]. Zhang et al. [14] reported an association between high SLC7A11 expression and advanced pathological differentiation in HCC tissues. Moreover, high SLC7A11 expression was associated with a shorter survival time, as compared with low SLC7A11 expression in HCC. In addition, SLC7A11 expression was reported to have a positive correlation with clinical data including lymph node metastasis status and disease recurrence status in tissues of colorectal cancer patients [15]. In concordance with our study, it was demonstrated that high SLC7A11 expression was significantly associated with the tumor growth type in CCA.

(line 401-403)

Similarly, CHAC1 overexpression in breast cancer tissues is associated with lymph node metastasis and increased cell proliferation, indicating poor prognosis in breast cancer [16].

(line 417-419)

However, the correlation between the expression of ACSL4 and SLC7A11 in sera and cancer patients’ prognosis is still less explored in publications and remains controversial in several diseases.

Point 3: Author should not only summarize the findings in the conclusion section but also suggest specific future research directions.

Author’s response: We appreciate and agree to the Reviewer’s suggestion. We have added future research directions in Conclusions (line 452-455) as following:

           (line 452-455)

This study provides good basic information for further study. In future research, the sample size should be expanded to investigate the role of ACSL4 and SLC7A11 and their inhibitors as a targeted therapy for improved therapeutic planning and management in patients with CCA.

Point 4: Again read the whole manuscript to correct the grammatical mistakes. Improvement in the grammatical mistakes will enhance the readability and professionalism of the article.

Author’s response: We appreciate the Reviewer's concern. The languages in our manuscript have been edited by Author Services Language Editing for MDPI. The certificate as below, and we have already rechecked the manuscript.

Point 5: Why has author mentioned some digits in bold text in Table 1, 2, 3, 4? If this indicates some significance, then it should be clearly mentioned in the table legend.

Author’s response: We thank the Reviewer’s concern. We have added the description of bold text “Statistically significant p-values are shown in bold.” in footnote of all tables according to the Reviewer recommendation.

Point 6: In line 361, “Zhang et al” was used, whereas “Mehta et al” was used in line 382. Correct them.

Author’s response: We acknowledge the Reviewer’s suggestion. We have made formatting changes to “Zhang et al” in line 376.

Round 2

Reviewer 1 Report

Comments and Suggestions for Authors

The revised manuscript has addressed the reviewers' comments effectively, demonstrating a clear effort to improve the quality and depth of the study. The authors have provided additional data, clarified methodological concerns, and acknowledged the study's limitations, particularly regarding sample size and the need for further research.

The inclusion of non-tumor bile duct tissues in the analysis strengthens the study's validity, mitigating concerns about the absence of a control group. The authors have appropriately acknowledged the speculative nature of the association between ACSL4 and ALT levels while suggesting it as a potential area for future investigation. This approach adds value to the study by proposing new research directions.

The statistical power calculations bolster confidence in the study's findings despite the small sample size. Additionally, the authors have provided satisfactory justifications for the antibodies used and have taken steps to validate their specificity, which enhances the reliability of the results.

Although the novelty of the work was questioned, the revisions have improved the manuscript substantially, addressing major concerns raised by the reviewers. The study's contributions to the field, particularly in proposing new biomarkers for cholangiocarcinoma, are of sufficient significance to merit acceptance.

Comments on the Quality of English Language

Minor editing of English language required.

Author Response

Reviewer 1 comments

Point 1:    The revised manuscript has addressed the reviewers' comments effectively, demonstrating a clear effort to improve the quality and depth of the study. The authors have provided additional data, clarified methodological concerns, and acknowledged the study's limitations, particularly regarding sample size and the need for further research.

                 The inclusion of non-tumor bile duct tissues in the analysis strengthens the study's validity, mitigating concerns about the absence of a control group. The authors have appropriately acknowledged the speculative nature of the association between ACSL4 and ALT levels while suggesting it as a potential area for future investigation. This approach adds value to the study by proposing new research directions.

                 The statistical power calculations bolster confidence in the study's findings despite the small sample size. Additionally, the authors have provided satisfactory justifications for the antibodies used and have taken steps to validate their specificity, which enhances the reliability of the results.

                 Although the novelty of the work was questioned, the revisions have improved the manuscript substantially, addressing major concerns raised by the reviewers. The study's contributions to the field, particularly in proposing new biomarkers for cholangiocarcinoma, are of sufficient significance to merit acceptance.

Author’s response: We appreciate the valuable comments from the reviewer that our manuscript sufficient for acceptance to publish in this journal.

Point 2:    Minor editing of English language required.

Author’s response: We thank you for the Reviewer’s comment. The languages in our manuscript have been edited by Author Services Language Editing for MDPI and we have already rechecked the manuscript. The certificate as following,